# Knowledge, Attitudes, and Practices Related to Cervical Cancer Prevention and Screening among Female Pharmacy Students at a Public University in a Southern Region of Saudi Arabia

**DOI:** 10.3390/healthcare11202798

**Published:** 2023-10-22

**Authors:** Vigneshwaran Easwaran, Eman Mohammed Shorog, Almaha Ali Alshahrani, Asif Ansari Shaik Mohammad, Mantargi Mohammad Jaffar Sadiq, Sirajudeen Shaik Alavudeen, Noohu Abdulla Khan, Md Sayeed Akhtar, Tahani Musleh Almeleebia, Sultan Mohammed Alshahrani

**Affiliations:** 1Department of Clinical Pharmacy, College of Pharmacy, King Khalid University, Abha 61421, Saudi Arabia; eshorog@kku.edu.sa (E.M.S.); sshaik@kku.edu.sa (S.S.A.); nakhan@kku.edu.sa (N.A.K.); mdhusain@kku.edu.sa (M.S.A.); shahrani@kku.edu.sa (S.M.A.); 2Department of Pharmacology, Pharmacy Program, Batterjee Medical College, Jeddah 21442, Saudi Arabia; jaffar.sadiq@bmc.edu.sa

**Keywords:** cervical cancer, screening, vaccination, prevention, students, awareness

## Abstract

Despite the availability of human papillomavirus (HPV) vaccines and screening facilities at various health centers in Saudi Arabia, the annual death rate due to cervical cancer is high. Therefore, knowledge and awareness are essential for self-care and educating others, particularly among healthcare students. The present descriptive, cross-sectional study explored female pharmacy students’ knowledge, attitudes, and practices related to cervical cancer. A total of 140 students participated in the survey. The survey was conducted for the period between April 2022 to September 2023. We observed a good knowledge score and positive attitudes among 8.5% and 93.5% of participants, respectively. A total of 10% of the study participants reported good practice scores. Most participants had never been screened for cervical cancer (94.3%). Among the non-screened subjects, feeling healthy and lacking information were the participants’ significant reasons for not screening for cervical cancer. A positive history of cancer related to smoking significantly impacted the knowledge score (*p* = 0.050). The current study reveals that healthcare awareness programs for cervical cancer and HPV vaccination are necessary at the level of educational institutions to improve public health.

## 1. Introduction

Cervical cancer is preventable and curable if diagnosed early [1]. More than 90% of cervical cancer cases are diagnosed in low and middle-income countries with a lack of organized screening and vaccination programs [2,3]. The introduction and implementation of Pap smear tests have significantly reduced mortality in developing countries [4]. Women’s use of cervical cancer screening varies across countries [5]. Cancer prevention strategies suggest that communicative programs should focus on primary cancer prevention, which includes multidisciplinary interventions, such as those promoting information and awareness, especially among young women, to increase the coverage of human papillomavirus (HPV) vaccination and promote screening services [6].

Most women usually do not experience symptoms related to cervical cancer until it becomes advanced, thus contributing to the disease’s advanced prognosis [5]. One of the most common reasons for the prognosis of cervical cancer is late presentation to a clinic. Late presentation to a cancer clinic makes treatment difficult and reduces the chances of successful recovery among cervical cancer patients [7,8]. This late presentation might be due to a lack of appropriate awareness [9]. Scientific research has proven that significant knowledge and awareness of cervical cancer improves preventive practice, HPV vaccine coverage, and the utilization of cervical cancer screening programs [10].

Cervical cancer is ranked as the fourth leading cancer among women worldwide and is ninth among Saudi women [1,11]. The Saudi MoH published guidelines for screening for HPV in 2014 and additionally made the HPV vaccine available for free to the public [12]. Despite the availability of HPV vaccines and screening facilities at various health centers, 358 women are diagnosed with cervical cancer, and 179 lose their lives annually [13,14]. This fact raises concern about women’s awareness and knowledge of cervical cancer in Saudi Arabia. Various awareness campaigns have been initiated but are limited to large cities. Additionally, even after the awareness campaigns, Saudi women doubt the value of HPV vaccination [15].

A great deal of knowledge, attitude, and practice (KAP) studies regarding cervical cancer have been conducted among various populations in various geographical settings in Saudi Arabia, and shockingly, some of the studies reported poor knowledge and awareness even among teachers, well-educated employees, and health workers [12,16,17,18]. Most of these studies were conducted among the general female population.

It is equally important to understand the level of knowledge and awareness among young women, such as students, since they are responsible for self-care and educating others [7,19]. To our knowledge, studies conducted among Saudi students are scarce, especially among pharmacy students. As pharmacy students are a part of the healthcare team, they can actively engage in preventive and public health activities [20,21]. Thus, the current study explored the knowledge, attitudes, and practices related to cervical cancer and its screening and identified the associated factors among pharmacy students. 

## 2. Materials and Methods

### 2.1. Study Design

This study involved an online institution-based cross-sectional survey conducted among female pharmacy students at King Khalid University, Abha, in the southern region of Saudi Arabia. We invited all undergraduate female pharmacy students to participate in the current study. The non-probabilistic convenience sampling technique was used. Students who were unwilling to participate and who refused to agree with the electronic consent form were excluded from the study, as were introductory pharmacy students. Additionally, partial responses and responses from participants who had never heard about cervical cancer were excluded.

### 2.2. Study Tool

The questionnaire was prepared by referring to various works in the literature [16,22] that assessed knowledge, attitudes, and practices related to cervical cancer and screening. The initial questionnaire was prepared and sent to experts for review. After an initial assessment, a finalized version of the questionnaire was designed. The questionnaire comprised four sections, namely those assessing demographic characteristics (7 items), knowledge (9 items), attitudes (8 items), and practices (4 items). A pilot survey was conducted, and the internal consistency of the questionnaire was estimated by calculating Cronbach’s alpha. Based on the results of the pilot study, the required modifications were made to the questionnaire. The finalized questionnaire was prepared as a Google form in English, and the link was made available to students through social media platforms such as Twitter^®^, WhatsApp^®^, and Facebook^®^.

### 2.3. Data Collection

The data were collected for the period between April 2022 to September 2023. The data collection form (Google web link) was made available through social media platforms such as Twitter^®^, WhatsApp^®^, and Facebook^®^ via the instructor to the class leader at the end of lectures. After the lectures, the survey link was closed, and no further responses were accepted to avoid the possibility of response bias. We repeated a similar procedure for the various years of pharmacy graduation. 

We continued the survey to get the maximum number of responses by using the same procedure mentioned above. We orally instructed the students to avoid re-submitting responses if they had already submitted them.

### 2.4. Scorings

The knowledge scores were obtained by measuring the correct responses on a scale of 0–1. Each correct answer was provided with 1 point, and each wrong answer was provided with 0 points. For items with more than one correct answer, 1 point was given if the respondent selected more than one correct response. One correct response and/or response of “do not know responses” were considered wrong. The valid responses to the knowledge assessment were categorized as good (80–100%), moderate (79–60%), and poor (<60%) based on Bloom’s cutoff points. The total knowledge score was calculated by adding up all the knowledge items’ scores, ranging from 0 to 10.

The attitude scores were derived using eight Likert scale-based statements. The responses ranged from strongly disagree to strongly agree, and the scores ranged from 1 to 5. A score greater than or equal to 20 was considered a positive attitude, and a score less than 20 was considered a negative attitude toward cervical cancer. The total score for the domain attitude was derived by adding up the scores of all the items. The range of scores in this domain was 8–40.

Practices were assessed by using responses to items regarding the use of preventive measures and screening for cervical cancer in the past. At least one positive response to the items included was considered a good practice. 

### 2.5. Statistical Analysis

The statistical analysis was performed using the Statistical Package for the Social Sciences (SPSS 22.0) (IBM Corp., Armonk, NY, USA). Descriptive statistics such as the mean, standard deviation (SD), frequency, and proportion were used to represent the socio-demographic characteristics and KAP of the study population. The chi-square test was used to analyze the categorical variables and estimate the associations between responses and socio-demographic characteristics. *p* < 0.05 was considered statistically significant.

### 2.6. Ethical Considerations

This study was approved by the research ethics committee of King Khalid University, and the approval number was ECM#2022-108. An informed consent form was included at the beginning of the electronic survey questionnaire, and only participants who gave informed consent were allowed further access to the questionnaire and participation in the study.

## 3. Results

### 3.1. Characteristics of the Study Subjects

A total of 185 pharmacy students responded to the questionnaire. Among these, 24 (13%) participants had not heard about cervical cancer. After excluding the responses by applying the exclusion criteria mentioned in the description of the methodology, 140 responses were obtained. The ages ranged from 18 years to 28 years. The mean age of the study participants was found to be 19.77 ± 6.708. Most participants were unmarried (97.1%), and 76.4% and 94.3% of the students who participated in the present survey had no history of cancer or cervical cancer, respectively. Approximately 14.3% of the participants in the present study had a history of smoking (Table 1).

### 3.2. Knowledge of Cervical Cancer Screening and Prevention 

On average, 48.7% of the study participants responded correctly to all the knowledge items included in the survey. More than 80% of the study participants knew at least one preventive method for cervical cancer. It was found that many of the study participants did not know symptoms (52.9%), risk factors (48.6%), and preventive methods (49.3%) for cervical cancer. A high frequency of correct responses to the items in the knowledge domain, including “What is the cause of cervical cancer?” (67.1%), “How can someone with cervical cancer be treated?” (66.4%), and “Can cervical cancer be cured in its early stages?” (61.4%) was found. A low frequency of correct responses was found for the following items: “Which screening method do you know to detect cervical cancer?” (28.6%), “What is the best time for performing a Pap smear test?” (31.4%), and “Who should be screened?” (33.6%) (Table 2).

### 3.3. Attitude towards Cervical Cancer Screening and Prevention 

Nearly 47.9% of the participants in the present study strongly believed that the early detection of cervical cancer saves lives. A total of 25.7% of the study participants thought they would have a chance of getting cervical cancer. Most study participants agreed or strongly agreed that cervical cancer may lead to death (62.2%) and that women of any age can acquire cervical cancer (57.9%). A total of 28.6% of the study subjects were neutral (neither agreeing nor disagreeing) about undergoing cervical cancer screening. However, 71.5% stated that cervical cancer screening helps prevent cervical cancer. It was found that 60.7% of the study subjects believed that cervical cancer could be treated successfully (Table 3).

### 3.4. Practice towards Cervical Cancer Screening and Prevention

Most participants in the present study had never undergone cervical cancer screening (94.3%). Menstrual irregularity was found to be the major reason for cervical cancer screening among those who had undergone it. Routine health screening is another primary reason for cervical cancer screening. Among the reasons for not undergoing screening for cervical cancer, the study subjects stated that feeling healthy was a primary reason (48.5%), followed by a lack of information about the screening methods (22.7%). Additionally, 93.6% of the subjects had not received a cervical cancer vaccine (Table 4).

### 3.5. Association of Demographic Characteristics with KAP Scores

The demographic characteristics were analyzed for their associations with the knowledge, attitude, and practice scores. A good knowledge score was observed among 12 subjects (8.5%). A positive attitude was observed among 131 subjects (93.5%). Good practices were reported by only 14 subjects (10%). None of the characteristics included in the present study affected the knowledge scores (*p* ≥ 0.05), except for smoking history (*p* = 0.05). Similarly, the attitude and practice scores were not influenced by any of the demographic characteristics included in the present study (*p* ≥ 0.05). The detailed scores and their statistical interpretation are provided in Table 5.

### 3.6. Predictive Factors of Likelihood of Receiving Cervical Cancer Screening and Cervical Cancer Vaccines

The present study evaluated possible factors that were predictive of whether the study participants underwent cervical cancer screenings and were likely to receive the cervical cancer vaccine (Table 6). The factors evaluated in the current study were not found to have a statistically significant impact on undergoing cervical cancer screenings or receiving cervical cancer vaccines.

## 4. Discussion

Early screening procedures and an effective vaccine have made cervical cancer an easily preventable disease [21]. The present study found inadequate knowledge of cervical cancer and its screening methods among the participants. These results are in concordance with those of various studies published around the world [18,23,24,25]. Although the participants in the present study had heard about cervical cancer, their knowledge was not up to the mark. This knowledge gap indicates the requirement for a nationwide awareness program for cervical cancer and its screening in the Kingdom of Saudi Arabia [24]. 

Compared to a study conducted among health and allied sciences students in the public university of Jeddah, our study participants’ knowledge of preventive measures and causative factors of cervical cancer was slightly higher. However, knowledge regarding the treatment and screening aspects of the disease was lower than that found in studies conducted in the central region of Saudi Arabia [11,26]. Similarly, to the findings in a report published by Zahid et al. [27], vaginal bleeding and painful intercourse were correctly reported as symptoms of cervical cancer by most of our study participants. Additionally, the lowest number of study participants considered smoking a risk factor. Unluckily, the participants in the present study lacked awareness regarding smoking, which is one of the risk factors for cervical cancer. However, most of them were smokers or had a history of smoking. 

This study also assessed the participants’ attitudes related to cervical cancer and its screening. The focus groups expressed no need for cervical cancer screening, and only half of the population was willing to receive the vaccine. These findings were inconsistent with those of studies published by Al Sairafi and Mohamed in Kuwait [28]. The reason behind this unwillingness might be a lack of information, lack of time, not wanting to undergo an invasive procedure, cultural and social factors, etc. [24]. Moreover, we identified that very few participants in our study had undergone screening for cervical cancer and received the cervical cancer vaccine. The rest believed that they were not at risk for developing cervical cancer. Similar attitudes and perceptions were observed among Saudi women from various regions [11,24,27,29]. The lower number of subjects screened for cervical cancer reflects the lack of awareness among pharmacy students, which is similar to that among common women. Many study participants also reported that cervix carcinoma leads to death, consistent with the results reported by Tadesse et al. One probable reason for this response is the widespread belief that all malignancies would result in death [30]. 

A study conducted among Ethiopian university students reported that only 2.2% of the participants had undergone screening for cervical cancer [9], which was comparatively higher than the rate in the current study and lower than in studies conducted in Hong Kong and Nepal [31,32]. The present study participants highly reported menstrual irregularity as a reason to undergo cervical cancer screening, followed by routine health check-ups. However, cervical cancer screening among our study participants was primarily hindered by factors such as feeling healthy and a lack of information. These results agreed with those of a study conducted by Tekle et al. [33]. Feeling shy about undergoing cervical cancer screening was another hindrance reported in our study and a similar study in Yemen [34]. These findings can be explained by poor knowledge and negative attitudes about preventing cervical cancer. A study conducted among female healthcare students in Malaysia reported a high rate of HPV vaccination, whereas it was low among our study subjects. This deficiency might be explained by their study’s participants’ heightened awareness in comparison with that of the present study’s participants [35].

The analysis revealed that no significant factors influencing the knowledge scores among pharmacy students existed. These findings are inconsistent with those of a study conducted in Ethiopia, where they reported that demographic characteristics were significantly associated with good knowledge scores and positive attitudes [33]. The present results revealed that a history of smoking was associated with poor knowledge. 

Various studies have established that a family history of cancer is a significant risk factor for cancer development [36,37]. Although a positive history of cancer among the respondents was not shown to have a statistically significant association, it somewhat influenced the attitude toward cervical cancer. Nonetheless, these results do not coincide with those of a population-based survey conducted in the southeastern United States [38]. A pooled analysis of a multicentric case-control study by the International Agency for Research on Cancer reported that smoking is positively associated with the prevalence of cervical cancer and increases the risk of cervical cancer, especially among human papillomavirus (HPV)-positive women [39]. Regrettably, most of the participants in the present study with a history of smoking were unaware of this. 

We could not find any characteristics in the current analysis that affected the nature of the study participants who underwent cervical cancer screening. These findings differed from those of other studies [15,40] in which the likelihood of undergoing cancer screening was proven statistically significant among those with a positive family history of cancer. 

It should be noted that a lack of awareness was reported not only among general students but also among healthcare students across the globe [26,41,42]. Therefore, awareness needs to be created among all students, including pharmacy and healthcare students. Studies conducted among Malaysian and Chinese students revealed that educational programs are required and effective in improving knowledge and awareness [23,25]. Therefore, awareness programs are the key to improving the health outcomes related to cervical cancer. The concept of salutogenic research suggests that the attention has shifted from pathogenic outcomes to public awareness, health promotion, and well-being. Along with awareness programs, health education campaigns shall be implemented [43]. Salutogenic principles such as abstinence from smoking, feminine hygiene, etc. shall be adopted in designing health education campaigns. If developed and designed effectively, health education campaigns can be considered rational as they increase knowledge, cognition, authorization, and understanding among the communities with individuals and help them consider various factors contributing to their well-being, develop resilience, and make predetermined informed choices for healthier lives. 

### Limitations

As a cross-sectional study, the present study had limited possibility of causal inference. The present study was conducted only among pharmacy students. Thus, the application of its results may not reflect the whole student population in the region or the entire Kingdom of Saudi Arabia. The sampling method could be another limitation that could lead to under or over-representation of the people. The low sample size in the current study might have produced a non-response bias. Agreement bias was possible due to the self-reported voluntary response.

## 5. Conclusions

The present study highlights the alarming ignorance, poor knowledge, and unfavorable attitudes related to cervical cancer among students, including false beliefs about the importance of vaccinations and smoking with respect to cervical cancer. However, some actions can be taken to raise student awareness of cervical cancer, vaccinations, and smoking.

First and foremost, educational initiatives centered on preventing, identifying, and treating cervical cancer should be created and implemented in schools and colleges. These initiatives should also include details on the advantages of vaccinations and the risks of smoking. Second, healthcare professionals might promote regular cervical cancer screening for girls and young women and tell them the benefits of being vaccinated against human papillomavirus (HPV), which is often the cause of cervical cancer.

Finally, to raise awareness about cervical cancer, vaccine adoption, and the dangers of smoking, particularly as it relates to cervical cancer, advocacy and prevention campaigns should be developed and distributed on social media platforms, within the community, and even through mainstream media. Effective add-ons such as the adoption of salutogenic principles can improve knowledge, cognition, and comprehension, helping people make healthier choices. To lower cervical cancer incidence and death rate, it is essential to increase student knowledge of the disease, the importance of HPV vaccinations, and the risks of smoking. 

## Figures and Tables

**Table 1 healthcare-11-02798-t001:** Characteristics of the study subjects.

Characteristics		Frequency	Percentage
Age (in years)	Range—18 to 28	Mean—19.77	* SD—6.708
Marital status	Single	136	97.1
	Married	3	2.1
	Divorced	1	0.7
Year of Pharmacy Education	Second year	31	22.1
	Third year	29	20.7
	Fourth-year	35	25
	Final year	45	32.1
Family history of cancer	Negative or Not known	107	76.4
	Positive	33	23.6
Family history of cervical cancer	Negative or Not known	132	94.3
	Positive	8	5.7
Smoking history	Negative	120	85.7
	Positive	20	14.3
	Total	119	100.0

* SD, Standard deviation.

**Table 2 healthcare-11-02798-t002:** Item-wise report on knowledge about cervical cancer, screening, and prevention.

Items Related to Knowledge	Correct Answer N (%)	Wrong Answer N (%)
1. What is the causative organism of cervical cancer?	94 (67.1)	46 (32.9)
2. What are the symptoms of cervical cancer?	66 (47.1)	74 (52.9)
3. Do you know the risk factors for cervical cancer?	72 (51.4)	68 (48.6)
4. How can an individual prevent cervical cancer?	71 (50.7)	69 (49.3)
5. Can cervical cancer be cured in its earliest stages?	86 (61.4)	54 (38.6)
6. How can someone with cervical cancer be treated?	93 (66.4)	47 (33.6)
7. Who should be screened?	47 (33.6)	93 (66.4)
8. Which screening method do you know to detect cervical cancer?	40 (28.6)	100 (71.4)
9. What is the best time for doing a Pap smear test?	44 (31.4)	96 (68.6)
Average of all items	68 (48.7)	72 (51.3)

N = Number of participants.

**Table 3 healthcare-11-02798-t003:** Item-wise report on attitudes about cervical cancer, screening, and prevention.

Items Related to Attitude	Strongly Disagree N (%)	Disagree N (%)	Neutral N (%)	Agree N (%)	Strongly Agree N (%)
Cervical cancer is highly prevalent and is a leading cause of death among all malignancies in Saudi Arabia	9 (6.4)	11 (7.8)	61 (43.6)	47 (33.6)	12 (8.6)
2. Do you believe early detection of cervical cancer helps in better management or treatment?	7 (5)	0 (0)	15 (10.7)	51 (36.4)	67 (47.9)
3. Do you believe that you have a chance of getting cervical cancer?	22 (15.7)	26 (18.6)	56 (40)	28 (20)	8 (5.7)
4. Do you think cervical cancer may lead to death?	8 (5.7)	8 (5.7)	37 (26.4)	69 (49.3)	18 (12.9)
5. Do you think any woman of any age can acquire cervical cancer?	10 (7.1)	8 (5.7)	41 (29.3)	68 (48.6)	13 (9.3)
6. Do you think cervical cancer can be treated successfully?	11 (7.9)	7 (5.0)	37 (26.4)	61 (43.6)	24 (17.1)
7. Do you think screening helps in the prevention of cervical cancer?	9 (6.4)	0 (0)	31 (22.1)	60 (42.9)	40 (28.6)
8. Are you willing to go for cervical cancer screening?	12 (8.6)	14 (10)	40 (28.6)	52 (37.1)	22 (15.7)

N = Number of participants.

**Table 4 healthcare-11-02798-t004:** Item-wise report on the practice of cervical cancer, screening, and prevention.

Items Related to Practice	Responses	Frequency N (%)
1. Have you ever been screened for cervical cancer	Yes	8 (5.7)
	No	132 (94.3)
2. Reason for screening	I have menstrual irregularity	2 (25)
	Had some symptoms	1 (12.5)
	Just a routine health screening	2 (25)
	Others	3 (37.5)
3. Reason for not being screened	I feel shy	11 (8.3)
	I am healthy	64 (48.5)
	I have no information about screening	30 (22.7)
	No symptoms	12 (9.1)
	It may be painful	6 (4.5)
	No reasons or others	9 (6.8)
4. Do you receive a cervical cancer vaccine?	Yes	9 (6.4)
	No	131 (93.6)

N = Number of participants.

**Table 5 healthcare-11-02798-t005:** Association of different characteristics with knowledge, attitude, and practice scores.

Characteristics			Knowledge Category			Attitude Category		Practice Category	
		Total	Poor	Moderate	Good	Negative	Positive	Poor	Good
Year of Pharmacy Education	Second year	31	18	7	6	2	29	30	1
	Third year	29	18	8	3	0	29	24	5
	Fourth-year	35	27	6	2	3	32	31	4
	Final year	45	32	12	1	4	41	41	4
X^2^			8.866			2.712		3.411	
*p*-value			0.181			0.438		0.332	
Family history of cancer	Negative or Not known	107	75	21	11	9	98	97	10
	Positive	33	20	12	1	0	33	29	4
X^2^			4.879			2.966		0.216	
*p*-value			0.087			0.085		0.642	
Family history of cervical cancer	Negative or Not known	132	90	30	12	9	123	120	12
	Positive	8	5	3	0	0	8	6	2
X^2^			1.462			0.583		2.121	
*p*-value			0.482			0.445		0.145	
Smoking history	Negative	120	85	24	11	9	111	108	12
	Positive	20	10	9	1	0	20	18	2
X^2^			5.989			1.603		0.000	
*p*-value			0.050 *			0.205		1.000	
Total		140	95 (67.9%)	33 (23.6%)	12 (8.5%)	9 (6.5%)	131 (93.5%)	126 (90%)	14 (10%)

* *p* value ≤ 0.05 considered significant, chi-square test.

**Table 6 healthcare-11-02798-t006:** Predictive factors of likelihood of receiving cervical cancer screening and cervical cancer vaccines.

Characteristics		Likely to Attend Cervical Cancer Screening			Likely to Take Cervical Cancer Vaccine	
		Total	Negative	Positive	Negative	Positive
Level of education	Lower level of education	95	90	5	89	6
	Higher level of education	45	42	3	42	3
X^2^			0.112		0.006 *	
*p*-value			0.738		0.937	
Family history of cancer	Negative or Not known	107	102	5	100	7
	Positive	33	30	3	31	2
X^2^			0.914		0.010 *	
*p*-value			0.339		0.921	
Family history of cervical cancer	Negative or Not known	132	125	7	124	8
	Positive	8	7	1	7	1
X^2^			0.725		0.520	
*p*-value			0.394		0.471	
Smoking history	Negative	120	113	7	113	7
	Positive	20	19	1	18	2
X^2^			0.022 *		0.495	
*p*-value			0.882		0.482	
Total		140	132	8	131	9

* A *p*-value less than 0.05 is considered significant.

## Data Availability

Data from this study are available upon request from the corresponding author.

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
