# Peer review of "Knowledge, Attitudes, and Practices Related to Cervical Cancer Prevention and Screening among Female Pharmacy Students at a Public University in a Southern Region of Saudi Arabia"

_healthcare, 2023, doi:10.3390/healthcare11202798_

Round 1

Reviewer 1 Report

Dear Authors,

 I thank you all for your hard work. I find this paper interesting, but very limited. The most important limit, according to my opinion, is limiting the questions only to the “HPV” and not to “prevention STD, breast cancer” and also the questions selected have their limits. Some arguments are important in order to define a correct health education campaign. Even if nowadays talk about health education campaign is already a limitation! The most advanced research in public health, today, tries to work on salutogenic principles.

In this research, it would have been interesting to know what men know on this specific issue……. 

English needs a review.

Author Response

We, the authors, would like to thank you for reviewing this manuscript and for your valuable comments.

Reviewer 1:

I thank you all for your hard work. I find this paper interesting, but very limited.

Comment 1: The most important limit, according to my opinion, is limiting the questions only to the “HPV” and not to “prevention STD, breast cancer” and also the questions selected have their limits.

Reply 1: Basing up on the primary research question, focusing on the KAP of cervical cancer and its possible prevention through HPV was important, it's also valuable to explore related topics like STD prevention and breast cancer for a holistic understanding of healthcare management.

Each question's scope has its purpose which was developed based on the review of literature as cited in the text, but considering broader health discussions can offer a more comprehensive perspective

Comment 2: Some arguments are important in order to define a correct health education campaign. Even if nowadays talk about health education campaign is already a limitation! The most advanced research in public health, today, tries to work on salutogenic principles.

Reply 2: Indeed, certain arguments are crucial for shaping effective health education campaigns. While discussions about health education campaigns have their limitations, it's essential to embrace the evolving field of public health research, which increasingly focuses on salutogenic principles for promoting well-being. As suggested, a note regarding the salutogenic principles has been added in the discussion and conclusion section for the perusal of the reviewers.

Comment 3: In this research, it would have been interesting to know what men know on this specific issue……. 

Reply 3: Thank you for bringing up a valid point. In future research, including an exploration of men's knowledge on this specific issue would provide valuable insights and a more comprehensive understanding of the subject.

However, our study focused on female subjects because self-care is more important, in addition to professional care since women are prone to this disease. Additionally, as per the literature evidence, self-awareness and knowledge play a significant role in decision-making by women to undergo cervical cancer screening and take the cervical cancer vaccine.

Comment 4: English needs a review.

Reply 4: Extensive English editing was done by using Grammarly's premium version and by using MDPI language editing services.

Reviewer 2 Report

This manuscript deals with cervical cancer knowledge:

One of the limitation of the study is the sample size. Thus the study is more adapted for a short communication which requires some summarizing of the manuscript (and by reducing the number of tables).

The second (most important) limitation of the study is the statistical analyses that require an extensive revision. In fact, including those who did not hear about cervical cancer in the analysis is misleading (we cannot ask the respondent about the cause of cervical cancer if they did not hear of the disease for example. The same remark applies for all the other questions. Revise please.

Also some percentages exceed 100% (i.e: item 2 and 3 in table 3).

You should define what do level 1,2… mean.

You should summarize the discussion (by deleting the sub-titles which may be added in the results) and thus deleting the introducing sentences.

The manuscript requires an extensive revision for the English language.

The conclusion needs to be revised and summarized. 

Extensive revision required

Author Response

We authors would like to thank you for your dedication in reviewing his manuscript.

Comment 1: One of the limitation of the study is the sample size. Thus the study is more adapted for a short communication which requires some summarizing of the manuscript (and by reducing the number of tables).

Reply 1: Despite the sample size limitation, the authors plan to present the data in a research communication rather than a short communication to ensure depth of analysis. We revised the manuscript accordingly

Comment 2: The second (most important) limitation of the study is the statistical analyses that require an extensive revision. In fact, including those who did not hear about cervical cancer in the analysis is misleading (we cannot ask the respondent about the cause of cervical cancer if they did not hear of the disease for example. The same remark applies for all the other questions. Revise please. Also some percentages exceed 100% (i.e: item 2 and 3 in table 3).

Reply 2: We value your input. To avoid misleading analyses, we will eliminate respondents who haven't heard of cervical cancer from pertinent questions. Therefore, it is required to revise the whole statistical analysis as recommended.

As recommended, the statistical analysis and the relevant tables were revised.

Comment 3: You should define what do level 1,2… mean.

Reply 3: The level of education was rewritten as a year of pharmacy education

Comment 4: You should summarize the discussion (by deleting the sub-titles which may be added in the results) and thus deleting the introducing sentences.

Reply 4: As suggested, all the subtitles and some of the introductory sentences were removed from the discussion section.

Comment 5: The manuscript requires an extensive revision for the English language.

Reply 5: The manuscript has undergone extensive language editing using Grammarly Premium. Additionally, the manuscript underwent language editing by MDPI language editing services.

Comment 6: The conclusion needs to be revised and summarized. 

Reply 6: As per suggestion, the conclusion was revised and summarized

Reviewer 3 Report

Thank you for the opportunity to review the manuscript. I have following suggestions:

§  In the abstract, there is a missing full stop at line 16.

§  The sample size of 119 is relatively small.

§  The introduction section requires restructuring as it currently contains a mix of ideas and lacks a clear flow. It should be divided into separate paragraphs, each emphasizing the necessity, rationale, and previous studies while connecting them to the global context and then to the Saudi Arabian context.

§  The data was collected through social media, and given the small sample size of 119, there may be biased findings. It is recommended to gather a larger sample size.

§  The presentation of the Odds Ratios in Table 6 and 7 is also not appropriate. Moreover, it raises the question of why Table 6 is included when there is no significant association among variables.

§  The manuscript requires significant English editing to address various typos and awkward phrasings. For instance, the word "Unfortunately" should be reworded in a more academic manner.

§  The paper still appears to be in a draft form and requires refinement in terms of overall organization, structure, and the use of English language.

§  The manuscript requires significant English editing to address various typos and awkward phrasings. For instance, the word "Unfortunately" should be reworded in a more academic manner.

Author Response

 Thank you for your kind review of this manuscript.

Comment 1:  In the abstract, there is a missing full stop at line 16.

Reply 1: As suggested, it was rectified

Comment 2:  The sample size of 119 is relatively small.

Reply 2: The sample size is one of the limitations, and it is added to the limitations section.

One potential explanation for the limited sample size is that the present study specifically targeted female pharmacy students only from a single university. In addition, the study has potential exclusion criteria for sampling.

Furthermore, in accordance with the recommendation of another reviewer, we incorporated an additional exclusion criterion into the research design. Consequently, further data was gathered in order to augment the sample size and afterward incorporated into the technique as phase 2 of the data-gathering process.

Comment 3:  The introduction section requires restructuring as it currently contains a mix of ideas and lacks a clear flow. It should be divided into separate paragraphs, each emphasizing the necessity, rationale, and previous studies while connecting them to the global context and then to the Saudi Arabian context.

Reply 3: As suggested, the introduction section was revised and restructured.

Comment 4:    The data was collected through social media, and given the small sample size of 119, there may be biased findings. It is recommended to gather a larger sample size.

Reply 4: It is important to acknowledge that the data collection process involves the utilization of social media platforms. However, it should be noted that the link to the survey was disseminated to the participants by the teacher, who shared it with the class leader at the end of the lecture sessions. Following the conclusion of the scheduled lectures, the survey link was subsequently deactivated, thereby precluding the acceptance of any more responses. This measure was implemented to mitigate the potential influence of response bias. The related statement was included in the methodology section.

Comment 5:   The presentation of the Odds Ratios in Table 6 and 7 is also not appropriate. Moreover, it raises the question of why Table 6 is included when there is no significant association among variables.

Reply 5: In response to a reviewer's suggestion, an extra exclusion criterion was incorporated into the study, prompting the administration of the survey once more to gather more data. Consequently, we conducted a subsequent statistical analysis by incorporating more data. Therefore, tables 5, and  6 were improved and table 7 was removed.

The display of data and its accompanying analytical information is deemed crucial for comprehending the concept, although it lacks statistical significance.

Comment 6:    The manuscript requires significant English editing to address various typos and awkward phrasings. For instance, the word "Unfortunately" should be reworded in a more academic manner.

Reply 6: The manuscript has undergone extensive language editing using Grammarly Premium. Additionally, the manuscript underwent language editing by MDPI language editing services.

Comment 7:    The paper still appears to be in a draft form and requires refinement in terms of overall organization, structure, and the use of English language.

Reply 7: The whole manuscript was revised and restructured.

The manuscript has undergone extensive language editing using Grammarly Premium. Additionally, the manuscript underwent language editing by MDPI language editing services.

Round 2

Reviewer 1 Report

Thank you for considering useful comments and suggestions

Author Response

No comments were received.

Thanks for your kind review and your input in improving the manuscript

Reviewer 2 Report

The authors should precise the study period in the abstract and in material and methods

The authors should justify why did the number participants has changed from the first manuscript.

The participants who didn't hear about cervical cancer are not to exclude. They should be included n the results but their knowledge should not be taken in consideration:

Table 4: correct the item 2: Reason for not being screened

Delete the response: "have a family history of cancer" since it is equal to zero

Table 6: correct as: Predictive factors of………..

Add subtitles to the results

Minor correction

Author Response

Comment 1: The authors should precise the study period in the abstract and in material and methods

Reply 1: As per suggestions, we have updated the abstract, and materials, and methods to include more detailed information regarding the study period. i.e., April’ 2022 to September’ 2023. As these changes will improve our study's readability and transparency and help readers understand better. We appreciate the reviewer's constructive criticism and dedication to revising the manuscript suitably.

Comment 2: The authors should justify why did the number participants has changed from the first manuscript.

Reply 2: Strict quality control methods taken during our research led to the modification of participants reflected in the revised text. During the initial submission of the manuscript, we included the participants who hadn’t heard about cervical cancer for the statistical analysis. However, after the first round of the review report, we reevaluated the inclusion and exclusion criteria to improve the quality of our study; subsequently, we excluded the participants who had not heard about cervical cancer for the final statistical analysis and it is shown in the revised manuscript.

Additionally, the data collection was continued during the revision phase of the manuscript in order to get the maximum number of responses; subsequently, the methodology was revised. Therefore, the number of participants changed from the first manuscript to the revised manuscript. Hence, we reanalyzed the data and revised the whole manuscript accordingly.

Comment 3: The participants who didn't hear about cervical cancer are not to exclude. They should be included n the results but their knowledge should not be taken in consideration.

Reply 3: We see the reviewer's point of view and agree that their input is valuable. As recommended by the reviewer, we believe that their answers should be included in the findings, along with a caution that their knowledge should not be incorporated into any conclusions or recommendations. Therefore, we did not include their responses for the final statistical analysis to avoid any possible misleading analyses since the study deals with knowledge, attitude, and practice. However, we mentioned the number and percentage of participants who didn’t hear about cervical cancer at the start of the results section. (Line no. 133 & 134)

Following the reviewer's advice, we adjusted the paper's methods and results sections to properly handle and report on data from these participants. We anticipate that this will strengthen our study's findings by making them stronger and more comprehensive.

Comment 4: Table 4: correct the item 2: Reason for not being screened

Reply: We believe it is item 3. As per the suggestions, it was changed in the re-revised manuscript

Comment 5: Delete the response: "have a family history of cancer" since it is equal to zero

Reply: As per the recommendations, the response “have a family history of cancer” was deleted in the re-revised manuscript

Comment 6: Table 6: correct as: Predictive factors of………..

Reply: Changed as per recommendations in the re-revised manuscript

Comment 7: Add subtitles to the results

Reply: Subtitles were added as per recommendations in re-revised manuscript

Comment 8: Minor correction on English

Reply: The whole manuscript was thoroughly checked and revised.